# A portable brightfield and fluorescence microscope toward automated malarial parasitemia quantification in thin blood smears

**Paul D. Gordon**[1]*, **Courtney De Ville**[2], **James C. Sacchettini**[2,3], **Gerard L. Coté**[1,4]

**1** Department of Biomedical Engineering, Texas A&M University, College Station, Texas, United States of America, **2** Department of Biochemistry & Biophysics, Texas A&M University, College Station, Texas, United States of America, **3** Department of Chemistry, Texas A&M University, College Station, Texas, United States of America, **4** Center for Remote Health Technologies and Systems, Texas A&M Engineering Experiment Station, College Station, Texas, United States of America

* gordonpd@ucmail.uc.edu

**Data Availability Statement:** All object count, classification, and statistical data can be found in the Texas Data Repository (https://doi.org/10.18738/T8/UBJTY1) in the form of MATLAB

## Abstract

Malaria is often most endemic in remote regions where diagnostic microscopy services are unavailable. In such regions, the use of rapid diagnostic tests fails to quantify parasitemia measurements which reflect the concentration of *Plasmodium* parasites in the bloodstream. Thus, novel diagnostic and monitoring technologies capable of providing such information could improve the quality of treatment, monitoring, and eradication efforts. A low-cost, portable microscope for gathering quantitative parasitemia data from fluorescently stained thin blood smears is presented. The system employs bimodal imaging using components optimized for cost savings, system robustness, and optical performance. The microscope is novel for its use of monochromatic visible illumination paired with a long working distance singlet aspheric objective lens that can image both traditionally mounted and cartridge-based blood smears. Eight dilutions of red blood cells containing laboratory cultured wild-type *P. falciparum* were used to create thin smears which were stained with SYBR Green-1 fluorescent dye. Two subsequent images are captured for each field-of-view, with brightfield images providing cell counts and fluorescence images providing parasite localization data. Results indicate the successful resolution of sub-micron sized parasites, and parasitemia measurements from the prototype microscope display linear correlation with measurements from a benchtop microscope with a limit of detection of 0.18 parasites per 100 red blood cells.

## Introduction

The aim of this study is to test a novel portable fluorescent microscope that may have utility for quantitative parasitemia measurement to inform malaria diagnosis. Despite improvements in global malaria infection and mortality rates over the past several decades, eradication

workspace and code used to process the data for publication.

**Funding:** This work was made possible by contributions from the National Science Foundation (NSF; https://nsf.gov/) in the form of research grant (#1402846; GC), graduate research fellowship position (GRFP #2017240275; PG), and the Precise Advanced Technologies and Health Systems for Underserved Populations (PATHS-UP) Engineering Research Center (1648451; GC) and by contributions from the Welch Foundation (https://www.welch1.org/) (grant A-0015; JS). The funders had no role in study design, data collection and analysis, decision to publish, or preparation of the manuscript.

**Competing interests:** The authors have declared that no competing interests exist.

progress has slowed since 2014, and the disease remains a global healthcare crisis with an estimated 228 million cases and 405,000 deaths annually [1]. Patients located in remote regions, which can be the most endemic for the disease, are often limited in their ability to receive the high-quality, laboratory microscopy diagnosis used in well-established hospitals and clinics [2, 3]. In the absence of better options, the presumptive, symptomatic-based diagnosis of malaria can have undesirably high inaccuracy and contribute to wasted resources, poor care, and the proliferation of antimalarial drug resistances [4, 5]. In the past decades, rapid diagnostic tests (RDT's) have been developed to screen for *Plasmodium* parasites in the blood with sensitivity at or below 200/μL, but they are unable to quantify parasitemia, or the concentration of parasites in the blood, which is used by clinicians to assess infection severity and monitor patient response to treatment. Recent guidance by the Malaria Eradication Research Agenda highlights the need for improved technological tools and data to guide diagnostic and eradication strategies in the coming decades [6].

Obtaining accurate parasitemia measurements is especially important for infections with *P. falciparum*, the deadliest and most common species of *Plasmodium* to infect humans, due to their ability to rapidly progress to complicated states [7]. Death from severe malaria often occurs within hours of admission for treatment, so it is essential that therapeutic doses of a highly effective antimalarial drug be administered quickly. Patients who are determined to be at a higher risk may be monitored more closely and placed onto more aggressive forms of treatment, whereas more conservative treatments may be prescribed for patients deemed to be low-risk [8]. While parasitemia alone is not an indication for disease severity, it can provide useful information to clinicians as they diagnose patients. High parasitemia in the absence of incapacitation can be an indicator that the disease may progress rapidly to a complicated state, especially in the presence of risk factors such as pregnancy, being below the age of 6–10, human immunodeficiency virus infection, malnutrition, or anemia [8]. This research aims to improve the quality of malaria diagnosis, treatment, and monitoring in remote regions by proposing and testing a low-cost, portable multimodal microscopy system.

Conventional parasitemia measurements collected according to the World Health Organization Malaria Microscopy Guide seek to provide a "reasonably and acceptably accurate" parasite count by comparing the ratio of parasites to leukocytes in thick blood films [9]. The accuracy of the method is hence highly variable and is limited by the assumptions that patients have 8,000 leukocytes and 5,000,000 red blood cells (RBC's) per microliter of blood. Many labs will take the additional step of verifying such parasitemia values against red blood cell counts obtained via hemocytometry. Values of parasitemia can also be defined by comparing the number of parasites to either red blood cells or directly to blood volume, which are also limited by their dependence on the precision of the assumptions of RBC's per microliter and the exact volume of blood sampled, respectively. To preserve clarity, this work will measure parasitemia by directly comparing the number of parasites to the number of red blood cells in a sample; either as the percent of infected RBC's or as an equivalent ratio of parasites per 100 RBC's, which is more descriptive of counts that include extracellular parasites in sample analysis.

In determining the desired lower bound of detection, it is impractical to use the true lower range of parasitemia found in patients, since it is sub-detectable using even conventional benchtop microscopy. Typical lower limits of microscopic detection from the literature range anywhere from 5–100 parasites per microliter, which is accomplished by examining thick blood smears. Approaching this same limit of detection is unfeasible using thin smear examination due to the high number of RBC's necessary to achieve statistical significance. It is most practical in this case to consider what lower bound of detection is implied by the diagnostic necessity when the microscope will be used as intended as a supplement to RDT screening. With the primary aim of disease diagnosis being to prescribe the proper course of treatment

for each patient, the system need only be able to quantify parasitemia insomuch as it assists clinicians in differentiating between uncomplicated cases, complicated cases, and those in danger of rapidly progressing into a complicated state. The microscope is most applicably useful, therefore, in identifying patients with high parasitemia but who do not present with obvious complicated symptoms. For this reason, the target measurement range for the system was decided to be between 0.01% - 1% of cells infected.

Taking microscopy out of a central laboratory for global health application is not a recent advancement in the field. In the 1930's-1950's, McArthur described the benefits of point-of-care brightfield microscopy [10–12]. A popular, recent trend leverages cell phone technology–their network access, processing power, onboard cameras, and other built-in sensors–to construct low cost, simple microscopes [13]. However, the ever-changing camera optics and small aperture lenses make their versatility, control, and quality assurance difficult to achieve [14]. In the time since MacArthur, various portable microscopes have been created and commercialized to varying degrees of success [15, 16]. In particular, the potential for fluorescence imaging to augment the signal-to-noise ratio (SNR) of portable microscopy has born several innovative diagnostic imaging systems, with each offering their own contributions to the progression of the field. Beginning in the 1990's, several systems presented various benefits and limitations to using fluorescence for malaria imaging [17–19]. During the next decade, the development of the CyScope® marked the first significant commercialization of a fluorescence microscope for malaria diagnostics in the field, although it relied heavily on manual operation and used conventional compound microscope optics [20–22]. In the past decade, the overall size of portable fluorescent systems has begun to shrink, utilizing new optical configurations, and their performance have begun to be evaluated in field settings [23–25]. More recently, the advent of machine-learning based image processing platforms has been extended to the interpretation of results from fluorescence scans of *Plasmodium* infected blood films [26].

## Microscope design and construction

### Microscope configuration

The microscope described here represents the completed third-generation system developed by the author's laboratory group, with several intermediary stages presented in conference proceedings [27–29]. Significant changes from the second-generation system include the use of lower cost optical components chosen by tuning magnification and resolution requirements, use of a higher-quality camera and more robust computational platform, and the elimination of cross-polarized imaging which was determined to add unnecessary cost and complexity. The microscope was designed to be bimodal, with brightfield imaging used to assess cellular size and boundaries and fluorescence imaging, which has been shown to be effective and less variable than Giemsa staining, used to detect intracellular parasites [23, 30]. The system was designed to resolve the features necessary to detect *Plasmodium* inside red blood cells from both conventional thin smears created on glass microscope slides and previously published thin smears generated using pumpless microfluidic cartridges [31]. To promote design simplicity, the center wavelength of the brightfield illumination source light emitting diode (LED) was set at 520 nm, in alignment with the fluorophore emission peaks of dyes used in the study. The use of a monochromatic system minimized the effects of chromatic aberrations and allowed for the use of singlet lenses where typically doublet or triplet lenses would be required for broadband systems. In order to resolve micron-sized features at $\lambda = 520$ nm, an objective numerical aperture of 0.40 was selected to provide theoretical resolution of 793 nm according to the Rayleigh criterion. Zemax OpticStudio ray tracing models were used to design the

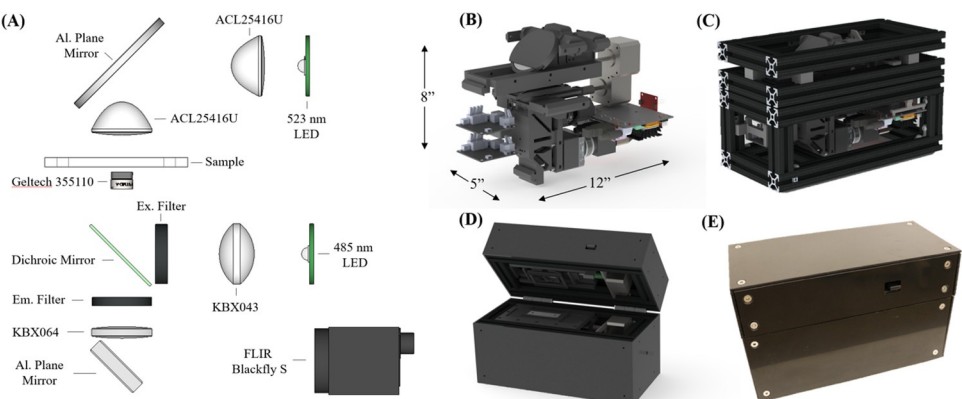

**Fig 1.** (A) Layout of optical components in the portable microscope. CAD models of microscope: (B) optomechanical mounts and electronic control components, (C) extruded aluminum frame, and (D) fully encased prototype demonstrating clamshell design. (E) Final physical prototype microscope.

illumination optics, infinity-corrected imaging system, choose off-the-shelf components, and verify performance prior to prototype construction. An overview of the microscope configuration can be seen in Fig 1.

## Prototype construction

The objective lens chosen based on modeling and experimental validation was a Geltech 0.40 numerical aperture (NA) glass aspheric singlet with 6.24 mm focal length (LightPath Technologies, Orlando, FL; #355110). A 1.6 MP monochromatic Blackfly S camera with a Sony IMX273 sensor (FLIR Systems Inc, Wilsonville, OR) was chosen for its combination of price, sensor and pixel sizes, frame rate, SNR, and well-defined programmatic interface. A bi-convex spherical singlet with 100 mm focal length (Newport Corp., Irvine, CA; #KBX064) was chosen as a tube lens, resulting in images being 16.03x magnified with a field-of-view (FOV) measuring 250 µm x 333 µm in the object plane, which allowed the full width of the previously cited microfluidic channels (250 µm) to be surveyed in a single image capture. A 523-nm monochromatic LED was chosen for transmission illumination (Osram, Germany; # LZ1-00G102-0000), and fluorescence excitation was provided by a 485 nm LED (Cree, U.S; #XPEBBL-L1-0000-00301). The fluorescence filter set (Semrock, U.S; FITC-LP01-Clinical-000) was chosen for compatibility with SYBR Green-1 dye that was used to label parasites in smears and contained a dichroic mirror with 500 nm cut on/off, 475/28 nm excitation bandpass filter, and emission long-pass filter with cutoff at 515 nm. Glass aspheric lenses with NA 0.79 were used to collect and condense the light from the transmission LED source onto the object plane (Thorlabs, Newton, NJ; #ACL25416U), and a singlet biconvex lens (Newport Corp, Irvine, CA; #KBX043) was used to collect epi-illumination light for fluorescence excitation. To reduce the footprint of the system, a two-inch diameter planar aluminum mirror (Thorlabs, Newton, NJ; #ME2-G01) was used to fold the brightfield illumination pathway and a one inch planar aluminum mirror (Thorlabs, Newton, NJ; #PFSQ10-03-G01) was used to fold the imaging pathway.

The prototype microscope was assembled using 3-D printed optomechanical components and was housed inside a case constructed from one-inch square aluminum rails and acrylonitrile butadiene styrene (ABS) plastic shrouds, with exterior dimensions measuring approximately 30 cm x 20 cm x 13 cm (Fig 1B–1E). The final device weighs approximately 6.5 lbs. The primary optical axis of the microscope was oriented vertically to allow samples to be held level

during imaging. The brightfield illumination system was housed in the lid of the microscope, and the imaging and control components were housed in the base. To operate, the lid was opened, a sample placed onto the stage, then the lid was closed before imaging. Interchangeable clips were used to hold either conventional 75 mm x 25mm glass slides or the aforementioned microfluidic cartridges, giving the microscope adaptability to either sample preparation format (Fig 2). With the lid closed, a compressive force is applied to the interchangeable clips, holding samples to the translation stage and ensuring alignment with the focal plane of the microscope. Sample translation and focal control utilize one stepper motor each (Sparkfun, Boulder, CO; # ROB-09238) to change fields-of-view and fine tune focus as needed. A Jetson Nano ARM-based single-board computer (Nvidia, Santa Clara, CA) was chosen as the onboard data collection and processing system due to its low cost and ability to facilitate future implementations of automated image processing based on machine learning algorithms. The system is currently powered using a dedicated power supply but can be adapted to operate from rechargeable battery packs for field deployment. A bill of materials for all major optical and electronic components is presented in S1 Table along with associated costs. The current prototype microscope system costs approximately $1,300, excluding the cost of non-critical components likely to be significantly changed during manufacturing such as fastening hardware, case rails, 3-D printed mounts, and wiring cables/adapters.

## Statistical analysis

A statistical analysis was conducted to determine the appropriate target parasitemia dilutions and inform the statistical significance of results depending on the sample size and density of parasites. A linear regression using Pearson's product moment correlation method was used to compare the values of parasitemia measured by the portable and benchtop fluorescent systems. If the determination of whether a specific red blood cell is infected is represented as a binomial event, the probability of whether red blood cells in a sample are infected with parasites can be assumed to follow a Bernoulli distribution. Given this assumption, it is possible to consider the relative percent error of a given parasitemia measurement at the edge of its 95% confidence interval for any given sample size. This relationship, being a function of two independent variables, can be represented as a surface over parasitemia and sample size, with the percent error

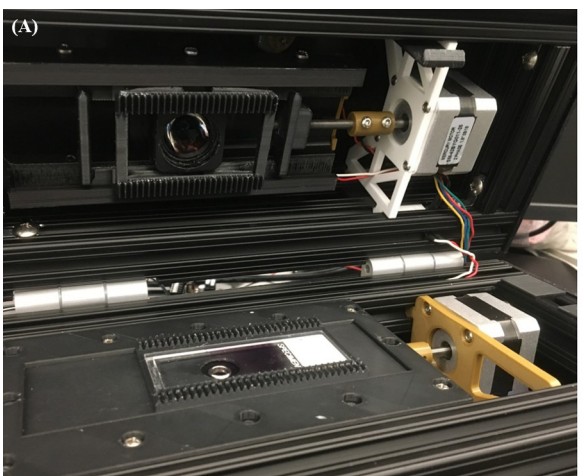
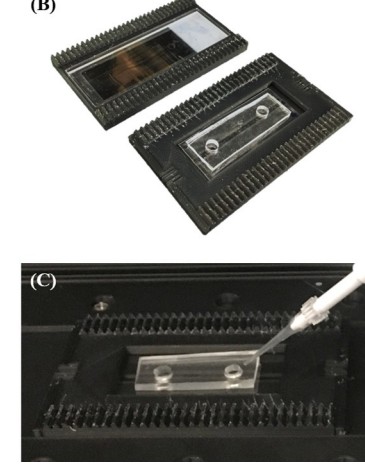

**Fig 2.** (A) A glass slide-mounted thin smear positioned in the open microscope, ready for imaging. The top motor assembly translates the sample while bottom motor controls objective focus. (B) Side-by-side comparison of clips for either glass or microfluidic smears. (C) Blood cells being pipetted into a microfluidic channel in the microscope.

of the confidence interval increasing with both lower parasitemia and lower sample size. A contour plot of this surface generated in MATLAB software using the Clopper-Pearson method is shown in S1 Fig, representing sample size as the equivalent number of microscopic FOV assuming the average cellular smear density (670 RBC per FOV) that was found using the bimodal microscopy system.

## Experimental procedure

*P. falciparum* wild-type strain 3D7 was obtained through BEI Resources (Manassas, VA; # MRA-102) and maintained in continuous in-vitro culture by a modification of the Trager and Jensen method [32]. AB+ human erythrocytes (Gulf Coast Regional Blood Center, Houston, TX) were used to suspend the parasites in a 4% hematocrit solution in complete medium containing 5 g Albumax II (Gibco Life Technologies, Auckland, NZ), 2 g sodium bicarbonate (EM Sciences, Gibbstown, NJ), 20 mg hypoxanthine (Sigma-Aldrich, St. Louis, MO), 16.2 g RPMI 1640 (contains 25 mM HEPES) (Gibco, Grand Island, NY), and 0.25 mL penstrep (Gibco, Grand Island, NY) per liter. Individual 5 mL cultures were grown in each well of a 6-well plate (Corning, Kennebunk, ME; # 3471) and incubated in a chamber flushed with a gas mixture of 2% $O_2$, 5% $CO_2$, 93% $N_2$ at 37˚C.

Wild-type *P. falciparum* parasites were cultured in cell media for nine days until erythrocytic life-cycle stages were desynchronized and multiple morphologies could be found simultaneously in test smears. From this culture, parasitized red blood cells were centrifuged in two-minute increments at 0.2 x 1,000 rcf to form a loose pellet of concentrated RBC's. These pellets were then extracted, collected into a single tube, and the process repeated twice again until cell media was separated from the parasitized RBC's. This concentrated culture pellet was resuspended in equal volume fresh human blood plasma to create a parasitized human blood analog. From images of Giemsa-stained control slides collected using 60x oil-immersion benchtop microscopy, the parasitemia of the raw culture was measured to be 1.2 ± 0.1% using digital counting of manually identified parasites and RBC's in ImageJ/FIJI software [33]. Various parasitemia dilutions were created based on the assumption of culture parasitemia equal to 1.2 parasites/100 RBC's (1.2%) by mixing volumes of this reconstituted parasitized culture with whole human blood. Dilutions were created to target parasitemia levels of 1% (50,000/μL), 0.75% (37,500/μL), 0.5% (25,000/μL), 0.25% (12,500/μL), 0.1% (5,000/μL), 0.05% (2,500/μL), 0.01% (500/μL), and un-parasitized blood (0.0% parasitemia). To promote the consistency of thin-smear quality, all data gathered for parasitemia quantification in this study used conventionally made smears on standard glass microscope slides. Fluorescent staining solution for dry thin smears was created by dissolving SYBR Green 1 stock at 1:8,000 dilution in 1x Tris-HCL buffer pH 8. To stain, previously fixed thin smears were flooded with 500 μL of SYBR stock solution and allowed to incubate for five minutes in the dark. Afterwards, smears were rinsed for 10 seconds with de-ionized water and allowed to air dry in the dark.

The portable microscope was connected to an auxiliary display monitor, keyboard, and mouse to control the user interface used to capture images, activate LED's, change exposure settings, and translate the sample. Alternatively, an iPad, laptop, or other device could be used to remotely operate the microscope through secure shell client software. Smears from each dilution were imaged first on the portable system, then on the benchtop system, with fluorescent images captured prior to brightfield imaging for each FOV to minimize the effects of photobleaching during brightfield image capture. Brightfield images were captured immediately afterwards, then the sample was translated to the next FOV and the process repeated until the usable monolayer region was exhausted. Manual inspection was used to identify whether or not a given FOV contained an even monolayer of cells, all FOV found to contain a

multilayer were bypassed by advancing to the following adjacent area on the slide and were thus excluded from data collection. This process could be automated in future control software development to increase the usability of the microscope by minimally trained personnel. In thin blood smears, cells tend to cluster more thickly on the proximal portions of the slide and thinly toward the distal regions, with a prime monolayer region tending to exist in between. To account for these heterogeneities in smear thickness, samples were initially mounted on the portable microscope stage with the FOV on one side of the monolayer region, and the slide was advanced linearly through the prime imaging area. Focusing of the objective was necessary before the first image was captured for each smear but re-focusing was typically unnecessary throughout the imaging of each individual slide. Due to the need to confirm proper focus of each FOV prior to imaging, all sample translation and focal adjustments are manually controlled using preset functions entered in the Python command terminal of the onboard Jetson Nano. Control of the responsible stepper motors could be easily automated with the development of auto-focusing functionality. After each smear was imaged on the portable system, the process was immediately repeated on a benchtop Nikon Ti-82 inverted fluorescence microscope using a similar commercial objective lens (20x, 0.45 NA, Nikon #MRH48230) and FITC fluorescence filter set, with fluorescence images again captured prior to brightfield images. The turnaround time for digitization of each FOV was not measured as a variable during experimentation. All data were saved from the portable microscope directly to a USB flash drive, and the uncompressed images were then transferred to a laptop computer for retrospective analysis.

### Ethical statement

This work represents early-stage technological and methodological development for potential medical diagnostic purposes. It has not yet been vetted in clinical studies and should not be used as guidance for medical diagnoses or treatments. The authors declare to have no financial interests or conflicts of interest in the technologies or methods herein described.

## Results

### Optical characterization

Brightfield images of a 1951 US Air Force Resolution Target (USAFT) (Fig 3A) and sections of sub-resolution fluorescent features (Fig 3C, inset) were used to derive the experimental resolution capabilities of the microscope. The system is fully capable of resolving sub-micron features, as shown by the visible distinction of all group nine USAFT elements (line widths = 0.98, 0.87, 0.78 μm), implying that the system approaches the theoretical diffraction limit of 0.793 μm for 0.40 NA imaging systems at $\lambda$ = 520 nm (Fig 3B). Further analysis of the US Air Force Target images shows that the net system magnification is experimentally measured to be 16.03x, matching the theoretical magnification predicted by the ratio of the tube lens focal distance ($f$ = 100 mm) to that of the objective lens ($f$ = 6.24 mm). At this magnification, each camera pixel represents 0.215 μm x 0.215 μm of object space, allowing the camera to satisfy the digital sampling requirements according to the Nyquist Sampling Theorem for features above 430 nm in size. The microscope's point-spread-function (PSF) was measured in sections through eight separate sub-resolution point features in fluorescence images (Fig 3C). After all normalized PSF's were averaged together, the full-width-half-max distance was calculated using linear interpolation to be 4.47 pixels, or 0.963 μm.

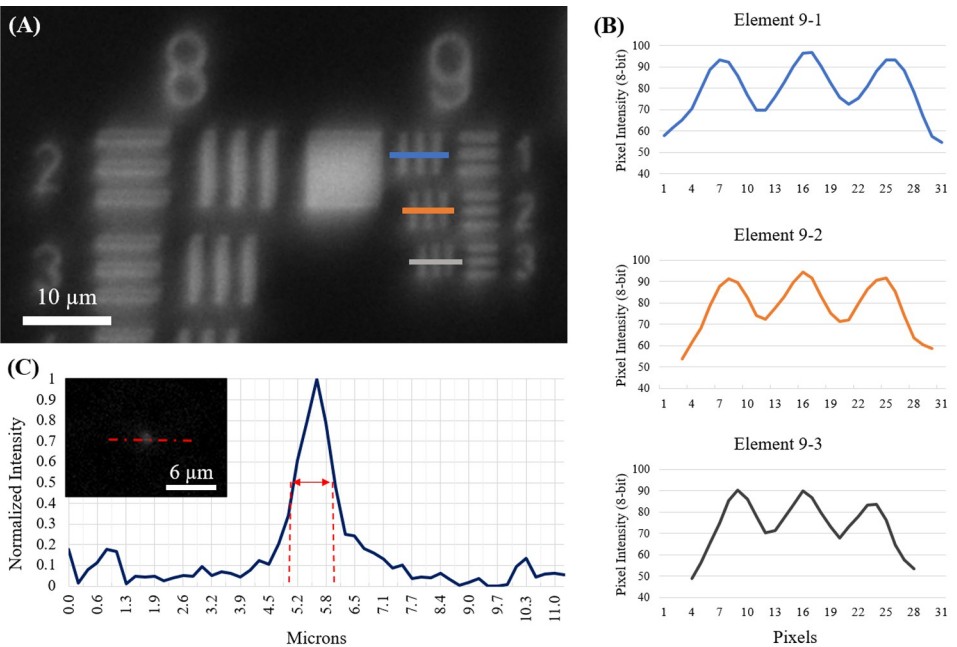

**Fig 3.** (A) Image of 1951 US Air Force Target captured using the portable microscope. (B) Profiles from rectangular sections taken across elements 9–1: 0.98 µm; 9–2: 0.87 µm; and 9–3: 0.78 µm are shown at right. (C) Point spread function of the microscope measured using averaged profiles through sub-resolution fluorescent features (inset).

### Quantitative parasitemia measurement

Thin blood smears used during investigation were measured to contain an average of 670 red blood cells per FOV on the portable microscope. When comparing brightfield images of the same sample gathered on both the prototype microscope and the benchtop control, the benchtop system showed more contrast at cell boundaries (Fig 4). Fluorescent images from the prototype system showed higher background fluorescence than their benchtop counterparts. Image exposure was tuned to be 100 ms for fluorescence images and 15 ms for brightfield images to prevent image saturation by maximally bright features for each modality. Subsequent brightfield and fluorescent images were collected on both systems with minimal pixel shifting between modes, allowing simple cross-comparison and image addition to create composite images that can be used to co-locate parasites and RBC's in a single frame.

Red blood cell and fluorescent objects in all images were counted using ImageJ/FIJI software, and data were processed using MATLAB software. The same object and RBC counting procedures were used for the generation of both the reference parasitemia measurements collected using the benchtop microscope and the experimental measurements from the portable system. All object data from images have been provided as a MATLAB workspace in the Texas Data Repository (https://doi.org/10.18738/T8/UBJTY1) along with the MATLAB script used to classify objects. RBC's per image were counted using the local maxima of the Laplacian transform of background-subtracted images, and parasites were counted in fluorescence images using a two-step segmentation procedure. Initial fluorescent object definition utilized automated intensity thresholding from background-subtracted images for segmentation, then final parasitic objects were separated from the population of all fluorescent features using thresholds set using the distributions of object maximum and mean pixel intensities, size, and circularity. The thresholds were set by the same investigator for all samples by using images from the raw culture smear as a baseline. All thresholds were verified against parameter

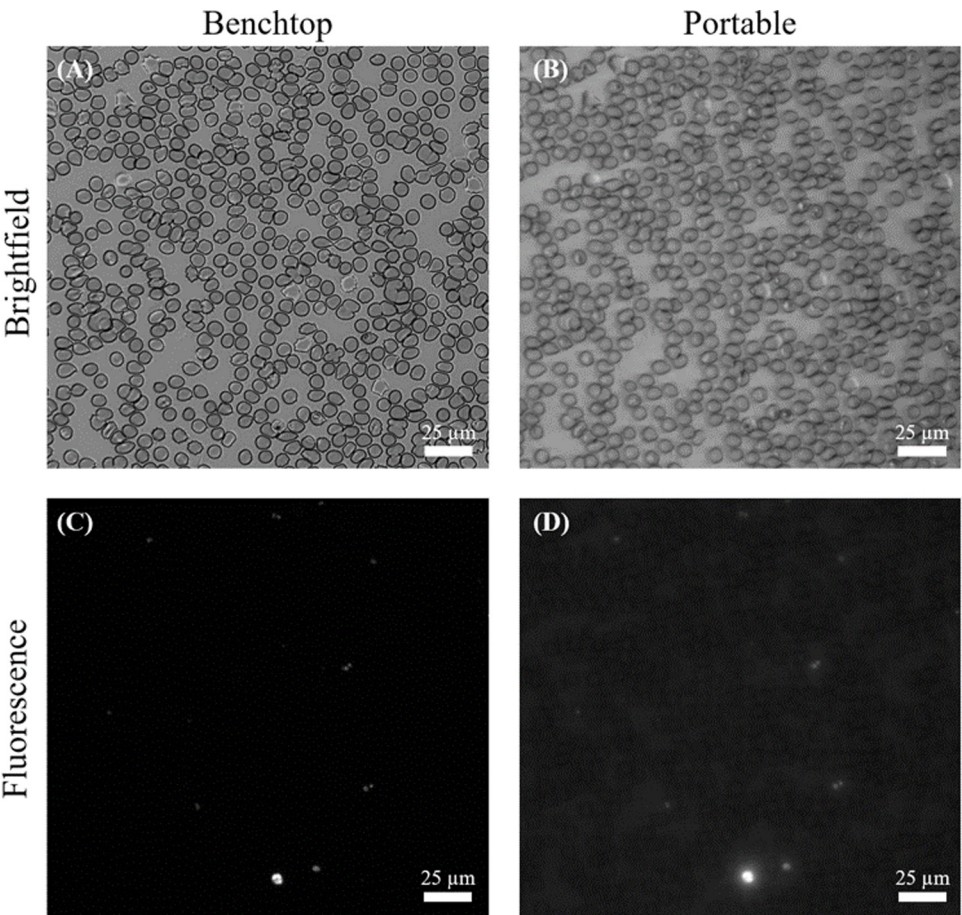

**Fig 4.** Brightfield (top) and fluorescence (bottom) images from the same FOV of Smear #3 for the benchtop control (left), and portable (right) microscopes. Images have been cropped and linearly contrast adjusted for clarity. Source benchtop images contain 4,275 cells and 44 parasites (1.0%) and source portable images contain 833 cells and 16 parasites (1.6%). The cropped images shown in the figure contain 593 cells.

distributions from manually segmented parasites used as classification controls. These thresholds were then automatically applied to all dilutions for image analysis. Object maximum pixel intensity was found to be the most discriminating feature for the differentiation of parasites and non-parasites as most fluorescent artifacts displayed peak fluorescence intensities below those of parasites.

After tuning the segmentation thresholds, the number of parasites and red blood cells from each field of view were tabulated, as shown in Table 1. From these data, the ratio of parasites to total red blood cells was plotted for the portable system and the benchtop system to assess the agreement in parasitemia measurements between the two systems (Fig 5). A linear regression was fit to the dataset to quantify the degree of agreement between the two sets of measurements, with adjusted $R^2$ = 0.939 and slope = 0.996. Parasitemia results are displayed as parasites per 100 RBC's to reflect the inclusion of all fluorescent objects determined to meet the criteria for being parasites, regardless of whether they were located inside or outside of blood cells. A comparison of images for high, medium, and low parasitemia from the portable microscopes can be found in S2 Fig.

**Table 1. Sample sizes of all smears, cells, and fluorescent objects.**

| Smear | Parasitemia Dilution | Portable Microscope | | | | Benchtop Microscope | | | |
|---|---|---|---|---|---|---|---|---|---|
| | | # FOV | # Cells | Raw Objects | Filtered Objects | # FOV | # Cells | Raw Objects | Filtered Objects |
| 1 | 1.200% | 21 | 12477 | 455 | 207 | 6 | 15441 | 754 | 244 |
| 2 | 1.000% | 26 | 13998 | 203 | 203 | 3 | 8247 | 107 | 107 |
| 3 | 1.000% | 19 | 12903 | 197 | 182 | 7 | 25817 | 406 | 294 |
| 4 | 0.750% | 50 | 30923 | 290 | 287 | 8 | 39139 | 215 | 215 |
| 5 | 0.500% | 10 | 7231 | 53 | 53 | 4 | 18703 | 98 | 98 |
| 6 | 0.500% | 25 | 17524 | 90 | 90 | 5 | 24216 | 111 | 111 |
| 7 | 0.250% | 9 | 6635 | 23 | 23 | 2 | 7045 | 43 | 16 |
| 8 | 0.250% | 24 | 14401 | 69 | 52 | 6 | 22843 | 103 | 88 |
| 9 | 0.100% | 20 | 15355 | 64 | 64 | 5 | 23089 | 43 | 43 |
| 10 | 0.100% | 17 | 11528 | 44 | 35 | 5 | 26413 | 45 | 42 |
| 11 | 0.050% | 25 | 22557 | 51 | 51 | 5 | 28322 | 32 | 15 |
| 12 | 0.050% | 13 | 9657 | 37 | 24 | 4 | 18215 | 33 | 33 |
| 13 | 0.010% | 35 | 18593 | 115 | 115 | 10 | 42366 | 85 | 85 |
| 14 | 0.010% | 32 | 24145 | 104 | 92 | 5 | 22662 | 64 | 64 |
| 15 | 0.000% | 18 | 13456 | 17 | 17 | 6 | 29521 | 4 | 4 |
| 16 | 0.000% | 10 | 6763 | 15 | 15 | 2 | 5248 | 9 | 2 |

## Discussion

### Advantages & novelty

Results of this early-stage study indicate that the prototype microscope can quantify parasitemia with high correlation to that provided by benchtop microscopy. The dynamic range of the system encompasses a range of parasitemia outcomes in which most high-risk infections might exist, enabling the likely identification of patients requiring urgent treatment when used in conjunction with rapid diagnostic screening tests. The system successfully uses several novel features for portable malaria microscopes–notably the use of a singlet aspheric objective lens capable of sub-micron resolution with a working distance of over 6mm. Use of this objective lens is possible due to the monochromatic illumination wavelength used which minimizes the presence of chromatic aberrations which would otherwise significantly distort images. Such a long working distance expands the flexibility of the system to image through additional types of samples and substrates, including conventionally mounted thin and thick blood smears and the cartridge-based smears previously mentioned. Typically, sample preparation is a major limitation for fluorescence microscopy because of the potential for artifact contamination, but the negative impact of staining contamination could be minimized by this microscope's compatibility with new, more self-contained sample preparation mechanisms.

To the author's knowledge, this work presents one of the lowest cost portable microscopy platforms capable of collecting and automatically processing quantitative parasitemia data for malaria patients at the point-of-care, although the eventual costs of commercially ready products are difficult to assess based only on preliminary prototypes. As a platform technology, the cost of the microscope should be divided across the usable lifetime of the device, allowing the net cost per test to be primarily driven by the cost of renewables (slides, cartridges, stains, etc.). Base model benchtop fluorescence microscopes retail for >$3,000 USD and these systems are not portable. Further, while simple compound benchtop microscopes can retail for $300–1,000 USD, these systems are also not generally portable nor capable of fluorescence imaging and do not have the onboard computational power required to provide the automation

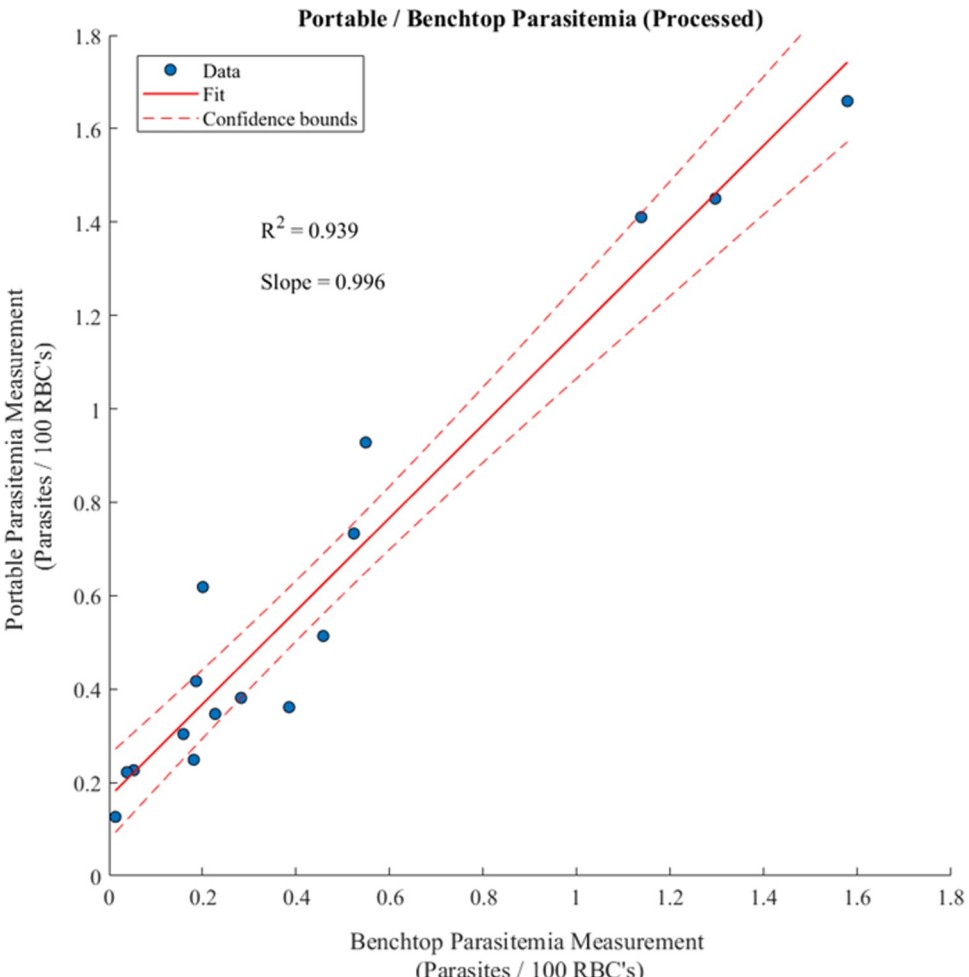

**Fig 5. Correlation between parasitemia measurements collected using the portable and benchtop microscopes.**
The fit line indicates a linear regression with reported $R^2$ value adjusted for the number of independent variables, and dashed lines bound the regression 95% confidence interval.

necessary to allow parasitemia quantification to be possible in remote settings by those without extensive skill and infrastructure. The prototype microscope presented here is comparable in price to previously published portable medical tools such as the Lab-In-A-Backpack system (~ $1,500) [34].

## Limitations

Several important limitations to the described microscopy system must be noted as well. This study should be considered to be early stage, as it demonstrates the theoretical feasibility of a novel optical microscopy system for use in malaria diagnostics. The study was conducted using a prototype system that is not commercially viable and did not examine samples from actual patients, which may display greater heterogeneity. Additionally, these experiments were not blinded and were conducted in a controlled laboratory without the presence of confounding environmental factors likely encountered in an actual use case. The harshness of field testing may necessitate design revisions such as kinematic features to ensure optical alignment of the illumination system when the lid is closed. One environmental challenge that will need to

be solved during field studies is the storage and transport requirements of the fluorescent stain. An alternative stain, Acridine Orange, may be substituted for SYBR Green-1 if a viable storage solution cannot be reached. Another limitation of this study is the lack of qPCR (quantitative polymerase chain reaction) use as a control standard for all microscopic measurements. When examining the results of the study, comparison of the intended parasitemia dilutions, which were generated using counts from traditionally Giemsa-stained samples, with the experimentally measured fluorescence values shows apparently higher numbers of parasites throughout in fluorescently stained samples. This discrepancy likely results both from A) the increased sensitivity of fluorescence staining to detect early stage and extracellular parasites that are not counted in traditional brightfield imaging and B) the inclusion of false-positive artifacts in the smear. Despite this, the benchtop and portable microscope proved to be comparable in their ability to detect fluorescent features in thin smears, demonstrating the performance of the portable optical system.

While displaying linear agreement with a benchtop equivalent, the data from using the portable microscope show a baseline false-positive rate of approximately 0.2 parasites per 100 RBC's, or 10,000 parasites per microliter. This lower limit of quantification should be sufficient to inform diagnostic decisions between complicated and uncomplicated cases, but it is limited to relatively high parasitemia cases and must be further improved for standalone use for confirmatory diagnoses. In its current state, the microscope must still be used primarily for parasitemia quantification as an adjunct to RDT-based parasite detection. It is expected that this lower limit of detection could be further improved using more sophisticated fluorescent feature classification that utilize the computational capabilities of the onboard Jetson Nano platform. Additionally, due to the fact that the number of FOV containing a monolayer of cells in a smear is dependent on the quality of the smear, there are inherent limitations to the possible number of cells to be surveyed, and thus the possible upper limit of accuracy, even if the entire smear is scanned. This dependency underscores the potential advantages of any sample preparation techniques that increase the quality of thin smears such that imaging and analysis can be conducted consistently.

## Conclusions

Results of this study show the imaging performance of a prototype microscope designed for the quantitative measurement of parasitemia in thin smears of *Plasmodium* infected blood. Quantitative parasitemia measurements have potential utility for monitoring disease severity both before and after the administration of treatment. The system demonstrates marked improvement in providing quantifiable data and compatibility with a variety of sample types compared to previous iterations of the system as earlier referenced. Further studies are needed, primarily in field-based settings, so that the diversity of possible hematological and contextual conditions can be encapsulated. Given its moderate-to-high limit of detection and the widespread adoption of RDT's in malaria diagnostic practices, it is anticipated that, once fully developed, this microscopy technique would be most effectively deployed as an adjunct to RDT's. In practice, the portable microscopy system could be utilized after *P. falciparum* infections are first confirmed using RDT's, creating a dual-technology method for efficiently screening and thoroughly diagnosing patients at the point of care.

## Supporting information

**S1 Fig. Relative measurement error.** Contour plot of relative measurement error at the edge of the measurement 95% confidence interval for varying parasitemia and sample sizes. Parasitemia is defined as the number of parasites per 100 red blood cells, and the sample size is listed

in number of fields of view examined assuming the average of 670 cells per field of view.
(TIF)

**S2 Fig. High, medium, and low parasitemia images.** Composite images from portable micro-scope showing smears with: (A) high 1% parasitemia, 833 cells, 13 parasites; (B) medium 0.1% parasitemia, 1005 cells, 4 parasites; (C) low 0.01% parasitemia, 833 cells, 2 parasites. The brightfield image from each was inverted and linearly contrast enhanced for clarity, and the fluorescent image overlay shows the segmented object pixels overlaid using the green image channel using FIJI ImageJ software.
(TIF)

**S1 Table. Costs of major components in portable microscope prototype.**
(PDF)

## Acknowledgments

The authors wish to thank Cody Lewis and Richard Horner for their assistance with the devel-opment of microscope control systems.

## Author Contributions

**Conceptualization:** Paul D. Gordon, Gerard L. Coté.

**Data curation:** Paul D. Gordon.

**Formal analysis:** Paul D. Gordon.

**Funding acquisition:** James C. Sacchettini.

**Investigation:** Paul D. Gordon.

**Methodology:** Paul D. Gordon.

**Project administration:** Gerard L. Coté.

**Resources:** Courtney De Ville, James C. Sacchettini, Gerard L. Coté.

**Supervision:** Gerard L. Coté.

**Writing – original draft:** Paul D. Gordon.

**Writing – review & editing:** Paul D. Gordon, Courtney De Ville, James C. Sacchettini, Gerard L. Coté.

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
