## [Decision Letter · Decision Letter 0]

14 May 2021

PONE-D-21-12786

A portable brightfield and fluorescence microscope toward automated malarial parasitemia quantification in thin blood smears

PLOS ONE

Dear Dr. Gordon,

Thank you for submitting your manuscript to PLOS ONE. After careful consideration, we feel that it has merit but does not fully meet PLOS ONE’s publication criteria as it currently stands. Therefore, we invite you to submit a revised version of the manuscript that addresses the points raised during the review process.

First, I would like to congratulate the authors for this interesting work. The manuscript is well written, it is clear and easy to follow. All the two reviewers have contributed tremendous feedback, with detailed analysis and valuable comments on how to improve the manuscript. Below there are several constructive comments that I encourage the authors to overcome and, I look forward to seeing the revised version. 

We look forward to receiving your revised manuscript.

Kind regards,

Érika Martins Braga, Ph.D.

Academic Editor

PLOS ONE

Journal Requirements:

Reviewers' comments:

Reviewer's Responses to Questions

**Comments to the Author**

1. Is the manuscript technically sound, and do the data support the conclusions?

Reviewer #1: Partly

Reviewer #2: Partly

2. Has the statistical analysis been performed appropriately and rigorously? 

Reviewer #1: I Don't Know

Reviewer #2: N/A

3. Have the authors made all data underlying the findings in their manuscript fully available?

Reviewer #1: Yes

Reviewer #2: Yes

4. Is the manuscript presented in an intelligible fashion and written in standard English?

Reviewer #1: Yes

Reviewer #2: Yes

5. Review Comments to the Author

Reviewer #1: PLOS ONE Review

Manuscript PONE-D-21-12786

Summary

The interesting manuscript entitled ”A portable brightfield and fluorescence microscope toward automated malarial parasitemia quantification in thin blood smears" by Paul David Gordon et al describes a digital system for P. falciparum malaria diagnostics where a low-cost, portable digital slide scanner is used to digitize fluorescently-stained blood smears, and the digital images analysed to detect and quantify P. falciparum parasites and red blood cells (RBCs). The system is evaluated by the analysis of thin blood smears, prepared from laboratory-prepared erythrocyte blood cultures with various levels of P. falciparum infection. Both the brightfield and fluorescent channels from each digitized field of view are analysed to detect parasites and RBCs, using a proprietary digital image algorithm. This digital analysis is performed based on segmentation of objects of interest (parasites in the fluorescent image and RBCs in the brightfield image) based on thresholds determined by manual analysis of the images. Overall, the results presented demonstrate a strong correlation in the quantification of parasites in the samples digitized by the proposed instrument, as compared to analysis of the same samples, digitized with a high-end reference digital microscope (Nikon Ti-82). As malaria remains a significant global health problem, novel technologies for improved field diagnostics are needed, and the technology presented here might therefore contribute to improving malaria diagnostics. The technology is aimed especially for point-of-care diagnostic utilization in low-resource areas where the burden of disease is typically the highest.

Overall, the technology presented by the authors is exciting, and the results encouraging. The quality of writing is very good, and the structure of the manuscript is generally quite good. Technical information regarding the hardware are presented in high detail, although some information regarding e.g. the development of the software is lacking. Although the findings are encouraging, and the research goal is laudable, I do however have several questions regarding the study methodology. Furthermore, although the results are presented as preliminary, I do believe some of the mains conclusions are not supported by the findings.

Comments:

1. Abstract - Please provide more information regarding study methodology in the Abstract. What types of samples were analysed and how were they prepared (i.e.thin smears, fluorescently stained, laboratory culture samples), how many samples were used, specific results (level of correlation) and reference standard used (visual or digital analysis of samples?). Also, avoid speculation in the Abstract which are not related to the research findings (e.g. here compatibility with machine learning solutions, this is not explored in the study).

2. Application and Potential Impact – I would consider moving this section (or the relevant information from this section to the Discussion)

3. Ethical statement - Although the submission information states that no ethical information is required, I would considering adding this information (as well as why no information is required) as a short statement to the manuscript body, as this is an article exploring a medical diagnostic technology.

4. I noticed your interesting previous work in this field from recent years (e.g. (https://spie.org/Publications/Proceedings/Paper/10.1117/12.2510593?print=2 and https://www.ncbi.nlm.nih.gov/pmc/articles/PMC6997630/)

• Is the same system used? Presumably these findings relate in some way, I would consider including a reference or brief summary on your previous work in this field. Have you updated the hardware, made technical adjustments or addressed other challenges?

5. Introduction (and Discussion) - Interestingly, we have published a study last year describing a very similar, also quite low-cost system with quite comparable imaging performance (https://journals.plos.org/plosone/article?id=10.1371/journal.pone.0242355). As you no doubt are aware, there are a number of other similar systems, as several other papers have investigated systems for field digitization of e.g. fluorescently stained samples (e.g. already in 2009 by Breslauer et al.), followed by e.g. CyScope and the Parasight system. I would consider briefly reviewing what the findings of previous work in the field have been to provide the reader with perspective and better understand the context and implications of your work. This is especially important as you state that this is the lowest cost similar system (the price seems to be approximately comparable to certain other solutions, although understandably it is difficult to establish a final price based on a prototype)

6. Page 6, line 116 - Please define abbreviation SNR (signal-to-noise ratio).

7. Line 151, page 8 – It seems Figure 2 and 3 have accidentally been interchanged (also e.g. Line 220, page 11)

8. Lines 187 – 201 - I would appreciate to read some more details about the operation of the system. How is the sample scanning performed – is the sample translated automatically or by manual adjustment of the sample to change FOV before digitization of the next one?

• How was the field of view (FOV) with the monolayer region selected?

• How large are the areas which are digitized (is the complete slide digitized)?

• Approximately how many RBCs and parasites were on average analyzed per slide (this is perhaps included in the MATLAB library online, but I would consider this information quite important to interpret the results, and therefore include some information also in the manuscript). If only a small image region is analyzed, the results might differ quite a lot based on the area which is analysed.

• Line 194 – what is an “adequate number of images” and how was this amount decided? Did it differ between samples?

• What was the turnaround time for digitization and analysis of one slide?

9. Methods and Results - How many thin smears did you analyze, i.e. what was the sample size and how was it decided? Was a single slide analyzed for each parasitemia dilution level? How were the statistical calculations performed, what was the level of significance and what software was used?

10. Methods - Although I understand that the parasitemia dilution levels were available from the preparation of the samples, these are apparently not used as a reference (e.g. Fig 5). It is a bit unclear to me what the reference standard is, e.g. is the Benchtop Parasitemia Measurement determined by manual (visual) analysis of the digital sample with the high-end microscope, or is the same digital image analysis software used, similarly to the low-cost solutions? If manual assessment of samples was used, how was the parasitemia level determined?

11. Methods - What samples were used to develop the software used for analysis of the samples, i.e. what samples did you use to determine the threshold settings to separate e.g. artefacts from parasites? Were these samples images from the same samples you analyzed, or completely different samples? Preferably the training data should be separated from the validation data.

• During our similar work, we found one unfortunately quite significant challenge to be the separation of artefacts from parasites by analysis of only the fluorescent samples, especially in conventional thin smears collected in the field (see e.g. S3 Fig https://journals.plos.org/plosone/article?id=10.1371/journal.pone.0242355#sec018, which represents a field sample). By brightfield analysis of Giemsa stained sample using high-power (60-100x) microscopy, it is possible to more reliably confirm the parasites, but e.g. the spatial resolution in Figure 4 seems to suggest that this would be quite challenging to do using the levels of magnification here. How did you confirm e.g. that the fluorescent signals with lower intensities that were classified as artefacts were not trophozoites? Although this is briefly discussed on page 12, perhaps e.g. image examples here could illustrate the difference between artefact and parasite more clearly.

12. Methods - How were digitized images stored – locally or uploaded to the internet? What is the file format and level of compression, and are the images saved as individual FOVs or stitched to whole-slide images? A figure demonstrating a “scanned sample” would perhaps illustrate this. Can the system operate in regions without network access?

13. Methods - As the system is aimed for rural, low-resource regions, can the system operate on battery power or does it require a stable power source? Naturally, this could likely of course be solved e.g. by using an external battery

14. Methods - Is the staining technique described here possible to perform in rural settings with access only to basic laboratory equipment, or would it need to be altered to be usable in low-resource environments?

15. Line 264, page 13 - “Preliminary results” - I would consider rephrasing to e.g. “early results” or clarify why these are preliminary. How do these results relate to your previous work (see earlier comments)?

16. Line 267, page 13 – As you have implicated, a detection level of 0.2 is quite low for clinical utilization and does not e.g. allow the ruling out infections. Please clarify how this is “sufficient to inform most diagnostic decisions” (i.e. in rural setting clinical malaria diagnostics, perhaps one of the most important clinical questions is the differential diagnostics of malaria vs other conditions).

17. Line 288, page 14 - Considering moving Table 1 to the Methods section (description of the system) or e.g. Supplementary Information and avoid presenting new information in the Discussion.

18. Line 290, p. 15 - “Results of this study show the feasibility of a method for quantitative parasitemia measurement while away from a centralized microscopy laboratory using an automated, low-cost bimodal microscope optimized for the specific task”. I do not agree that the results suggest this. The reason is that the sample are neither collected, nor prepared (to my understanding) while away from a centralized laboratory. Although the samples are thin smears, I would consider it quite unlikely to be directly comparable to thin blood smears collected in field settings. To my understanding, the results seem to suggest instead that the imaging performance of the presented system seems to be sufficient for determination of the parasitemia level in moderate- to high-level samples, which are prepared in laboratory settings. I therefore think it is premature to draw conclusions regarding usage especially at more remote points of care. I would consider rephrasing this to e.g. instead state that the system is designed for usage at the point of care.

19. Discussion – Please further discuss the strengths and weaknesses of the study, as well as the challenges required before clinical implementation. E.g. although it is stated that the technology could be combined with RDTs for diagnostic applications, it should be emphasized more clearly that the detection level is quite high and needs to be improved to allow confirmatory diagnostics.

20. Figure 1 – Please include size of the system in the image (width, length, height) and weight. This could be included in the Methods section also.

21. Figures - Although the data from all findings are made available, is any image data available (e.g. illustrating the detected parasites in e.g. a low-, compared to a high-level parasitemia sample? This would make the analysis of the samples more understandable also for non-technical readers, such as clinicians.

Reviewer #2: Summary of the research and overall impression

This well-written manuscript describes the results of a laboratory-based study to compare a newly developed portable malaria microscopy device with bench-top microscopy on blood samples spiked with cultured plasmodium falciparum parasites. The study shows that the new portable device is able to detect and quantify parasitemia to a similar degree as the bench-top microscope. The authors conclude that this technique could be used in combination with RDT’s to monitor treatment efficacy in malaria endemic settings and that it could contribute to screening for antimicrobial resistance.

The results section shows figures of the device and the images the device can produce compared to the images produced by a bench-top microscope. It is not entirely clear to me if the device provides a clear-cut interpretation of results, or if a technician is still needed to interpret the data, though the title of the manuscript suggests the former. It furthermore provides a linear regression model in which the quantification results from the bench-top microscope are compared to the portable microscope.

The manuscript furthermore provides a detailed insight into how the device was developed. The added value of the portable device over regular bench-top microscopy would be the fact that it is automated (I assume) and that it is (as the name suggests) portable, and therefore can be moved without having to be calibrated before use. If this technique is finetuned (a baseline false-positive of 10.000 p/uL is rather high) and works in a rural setting it could prove useful in malaria management.

The main issue of this study to me is the fact that the authors draw their conclusions based on a relatively small non-blinded laboratory-based experiment without testing the device in field settings and clinical samples.

- How do the microfluidic cartridges work (these would be a major advantage compared to the use of thin slides as to create a good thin slide requires some skill)

- A blinded study using spiked samples

- A blinded study using clinical samples

- Preferably a time-series before and after treatment

- Preferably a field study performed in field settings to see how it copes with differences in temperature, humidity, dust etc.

Second, I think the manuscript should focus more on the experiment itself and the results of the experiment. At the moment the only explicit data shown as to how it functions compared to other techniques is a figure to compare images made by bench-top microscope and the device, and one linear regression analysis.

This is why I suggested a major revision. Personally, I would like to see additional experiments done before these conclusions are drawn. If the authors would adjust the conclusions to better reflect the actual data presented here a minor revision would be justified. I think that the device has potential, but I also think that it's potential is very limitedly shown in the current data.

General Comments

- A baseline false-positive of 10.000 p/uL is a very high baseline, limiting the usefulness of this technique as it is presented at the moment, and should be acknowledged as such. Particularly taken into consideration that RDT’s have to be able to detect parasitemia at 200 p/uL and that field microscopy is generally able to detect parasitemia counts lower than that

- The experiment was not blinded which should be explicitly stated.

- Samples were not tested in duplicate

- Parasitemia levels in the dilutions were assumed rather than tested

Other comments

- The manuscript places emphasis on technical details on how the portable microscope was designed. As a result, it can be challenging to read for a broader audience.

- It would be useful if the authors would consistently provide both percentage infected RBC and p/uL or choose one of the two and leave out the other. Right now both are used interchangeably which makes it more difficult to compare

- To me it seems like there are a lot of methods in the results section and some new data (e.g. the cost overview as well as the statement that the LOQ was 10.000 p/uL) in the discussion which I found somewhat confusing, though it is a minor remark.

In addition to the comments mentioned above

Abstract

- In my opinion the authors should be careful with their hypothesis that the technology may be useful in drug resistance screening. First of all, the technique itself does not screen for resistance but rather assesses efficacy of therapy. Second, the current study does not provide data to support this claim as they did not test follow-up, nor any clinical samples. In this, the technique is no different from any other which can quantify parasitemia.

Introduction

- Line 57; low parasitemia counts do not necessarily correspond with mild disease and vice versa; I would nuance this statement.

- Line 66: I would nuance this by adding that in many labs data are corrected for bloodcell counts as obtained by hemocytometry.

- It would be useful to add the aim of the current study to the introduction (the current study describes …..) and not just the ultimate goal of the technique that is being developed (line 60 onward). This makes it easier for the public to screen what they are about to read which was a lot more technical than I had anticipated from the title and abstract.

Methods

- Please describe how many samples were tested and what their parasitemia level was according to the culture. Please state how these relate to the portable microscope results

- Please describe in more detail how test results were interpreted (automated or not) and who interpreted the results (same person for both tests?)

- The FOV without monolayer cells excluded: did this significantly decrease the number of fields included in the analysis? Please provide numbers on how many fields were captured per slide and how many were excluded. Please state in the discussion how this would affect the results

- I wonder if logistic regression comparing test results of two tests which were (as I assume) interpreted by thesame researcher is an appropriate choice. If possible, it would be nice to compare with qPCR instead.

Results

- It would be nice to see an image of how the results are presented to the technician.

Discussion

- I am missing a limitations section

6. PLOS authors have the option to publish the peer review history of their article (what does this mean?). If published, this will include your full peer review and any attached files.

Reviewer #1: **Yes: **Oscar Holmström, MD, PhD

Reviewer #2: No

---

## [Author Response · Author response to Decision Letter 0]

26 Oct 2021

The authors wish to sincerely thank the Reviewers for their excellent and thorough feedback, especially as it relates to clarifying the purpose of the work and incorporating it into the broader field. The authors also thank the editors and reviewers for their patience as revisions have been completed.

Overall, there have been many minor revisions that should improve the robustness of the work and the clarity of communication for the work that has been done. The primary changes were to emphasize that the work was meant to be a technical description of a new technology rather than a description of a diagnostic test itself. Nearly all comments were accepted and implemented in the paper.

---

## [Decision Letter · Decision Letter 1]

6 Dec 2021

PONE-D-21-12786R1A portable brightfield and fluorescence microscope toward automated malarial parasitemia quantification in thin blood smearsPLOS ONE

Dear Dr. Gordon,

Thank you for submitting your manuscript to PLOS ONE. After careful consideration, we feel that it has merit but does not fully meet PLOS ONE’s publication criteria as it currently stands. Therefore, we invite you to submit a revised version of the manuscript that addresses the points raised during the review process. Please address the comments and suggestions carefully and submit your revised manuscript by Jan 20 2022 11:59PM. If you will need more time than this to complete your revisions, please reply to this message or contact the journal office at plosone@plos.org. Please include the following items when submitting your revised manuscript:A rebuttal letter that responds to each point raised by the academic editor and reviewer(s). You should upload this letter as a separate file labeled 'Response to Reviewers'.A marked-up copy of your manuscript that highlights changes made to the original version. You should upload this as a separate file labeled 'Revised Manuscript with Track Changes'.An unmarked version of your revised paper without tracked changes. You should upload this as a separate file labeled 'Manuscript'.If applicable, we recommend that you deposit your laboratory protocols in protocols.io to enhance the reproducibility of your results. Protocols.io assigns your protocol its own identifier (DOI) so that it can be cited independently in the future. For instructions see: https://journals.plos.org/plosone/s/submission-guidelines#loc-laboratory-protocols. Additionally, PLOS ONE offers an option for publishing peer-reviewed Lab Protocol articles, which describe protocols hosted on protocols.io. Read more information on sharing protocols at https://plos.org/protocols?utm_medium=editorial-email&utm_source=authorletters&utm_campaign=protocols.

We look forward to receiving your revised manuscript.

Kind regards,

Ming Dao, Ph.D.

Academic Editor

PLOS ONE

Journal Requirements:

Reviewers' comments:

Reviewer's Responses to Questions

**Comments to the Author**

1. If the authors have adequately addressed your comments raised in a previous round of review and you feel that this manuscript is now acceptable for publication, you may indicate that here to bypass the “Comments to the Author” section, enter your conflict of interest statement in the “Confidential to Editor” section, and submit your "Accept" recommendation.

Reviewer #3: All comments have been addressed

Reviewer #4: (No Response)

2. Is the manuscript technically sound, and do the data support the conclusions?

Reviewer #3: Yes

Reviewer #4: Yes

3. Has the statistical analysis been performed appropriately and rigorously? 

Reviewer #3: N/A

Reviewer #4: Yes

4. Have the authors made all data underlying the findings in their manuscript fully available?

Reviewer #3: Yes

Reviewer #4: Yes

5. Is the manuscript presented in an intelligible fashion and written in standard English?

Reviewer #3: Yes

Reviewer #4: Yes

6. Review Comments to the Author

Reviewer #3: I thank the authors for their great effort to address the comments from the reviewers. The revised manuscript has been greatly improved from the first draft.

Two more comments:

1. The units standard used by the authors are not consistent. In Line 164~165, I encourage the authors to change to unit to international standard unit, i.e., mm and kg.

2. The illumination system of the microscope was housed in the lid of the microscope. When the lid was opened and then closed, will the alignment of the lid with the base affect the image quality? I suggest to add some mechanism along the edge of the lid and base (some sort of kinematic design) to ensure better alignment.

Reviewer #4: This resubmission by Paul Gordon et al. describes the development of a low-cost, portable microscope with brightfield and fluorescence imaging functions. This bimodal imaging capability is helpful in detection of Plasmodium parasites in thin blood smears using relatively low magnification lens. Design and engineering of the microscope prototype are described in great details. The strengths of this study are the novel bimodal microscopy, low cost ~ $1,300, and its compatibility to blood smears prepared on conventional glass slides and inside microfluidic channels. Overall, this paper has its merit, and the writing is fluent and generally clear. Some questions should be clarified by the authors as listed below:

Comments:

1. Was there any attempt to confirm the parasitemia of the Plasmodium cultures using benchtop microscope under higher magnifications? For example, immersion oil 60x or 100x can provide accurate analysis on parasites through early ring to late trophozoite stages. If so, it would help validate the measurements while eliminate the requirement of PCR use and reduce the effects of the possible artifacts of the fluorescence microscopy.

2. Sample size (number of FOVs) is dependent on the quality of the smears, where only the FOVs of monolayer of cells is collected and analyzed. Although the preliminary results using dilutions of parasite cultures show good agreement between benchtop and portable measurements at high parasitemia, >500 parasites/µl blood. This may be problematic for the intended field use and PoC settings, for real samples with low parasitemia and poor-quality smears.

3. It is unclear what algorithm was used to differentiate monolayers from multilayers. Was it done manually during microscopy and data collection or retrospectively?

4. The present imaging analysis method does count all parasites in the smears. Could an algorithm be added to correlate between brightfield and fluorescence images of a same FOV, so that cellular and extracellular parasites can be differentiated?

5. (minor) Authors removed all relevant descriptions and claims on drug resistance testing. In my opinion, a significant advantage of microscopy method than RDTs in malaria detection is that it can be potentially useful in differentiation live parasites from dead parasites for drug resistance screening. Can you comment would it be possible to implement such function, e.g., addition of another fluorescence filter? If so, what would be the extra amount of work/cost? I understand the current revision has been made to focus more on the diagnostic applications.

6. Fig. 4, please add quantifications of these representative images, e.g., RBC count and parasites count analyzed from both benchtop and portable microscope images.

7. I would appreciate some representative image data of low, moderate, and high parasitemia samples being added, either to the Fig. 4 or in the supporting information.

Other comments:

1. Line 310 has a typo “hve”.

7. PLOS authors have the option to publish the peer review history of their article (what does this mean?). If published, this will include your full peer review and any attached files.

Reviewer #3: No

Reviewer #4: No

---

## [Author Response · Author response to Decision Letter 1]

26 Feb 2022

Thank you again for your excellent feedback. The comments from both reviewers were entirely appropriate and appreciated. 

With specific regards to the comments on the potential for live parasite imaging, although it is not a part of this work, the separate publications on micro-fluidic channel smears hold some promise in this regard, although the technology needs further improvements to be deployable. We welcome further feedback and discussion on this topic in the future as appropriate.

---

## [Decision Letter · Decision Letter 2]

22 Mar 2022

A portable brightfield and fluorescence microscope toward automated malarial parasitemia quantification in thin blood smears

PONE-D-21-12786R2

Dear Dr. Gordon,

We’re pleased to inform you that your manuscript has been judged scientifically suitable for publication and will be formally accepted for publication once it meets all outstanding technical requirements.

Kind regards,

Ming Dao, Ph.D.

Academic Editor

PLOS ONE

Reviewers' comments:

Reviewer's Responses to Questions

**Comments to the Author**

1. If the authors have adequately addressed your comments raised in a previous round of review and you feel that this manuscript is now acceptable for publication, you may indicate that here to bypass the “Comments to the Author” section, enter your conflict of interest statement in the “Confidential to Editor” section, and submit your "Accept" recommendation.

Reviewer #3: All comments have been addressed

Reviewer #4: All comments have been addressed

2. Is the manuscript technically sound, and do the data support the conclusions?

Reviewer #3: Yes

Reviewer #4: Yes

3. Has the statistical analysis been performed appropriately and rigorously? 

Reviewer #3: I Don't Know

Reviewer #4: Yes

4. Have the authors made all data underlying the findings in their manuscript fully available?

Reviewer #3: Yes

Reviewer #4: Yes

5. Is the manuscript presented in an intelligible fashion and written in standard English?

Reviewer #3: Yes

Reviewer #4: Yes

6. Review Comments to the Author

Reviewer #3: I thank the authors' effort in improving this manuscript, which is ready to be published. I found the manuscript easy to read and concise. I just have one more suggestion to the authors, in line 165, can the authors change 6.5lbs to kg, just to be consistent with other units?

Reviewer #4: All comments have been addressed well. The writing is fluent and results are clearly presented. I would recommend for acceptance.

7. PLOS authors have the option to publish the peer review history of their article (what does this mean?). If published, this will include your full peer review and any attached files.

Reviewer #3: No

Reviewer #4: No

---

## [Editor Report · Acceptance letter]

30 Mar 2022

PONE-D-21-12786R2 

A portable brightfield and fluorescence microscope toward automated malarial parasitemia quantification in thin blood smears 

Dear Dr. Gordon:

I'm pleased to inform you that your manuscript has been deemed suitable for publication in PLOS ONE. Congratulations! Your manuscript is now with our production department. 

Kind regards, 

on behalf of

Dr. Ming Dao 

Academic Editor

PLOS ONE